# TEMPORAL PROBABILISTIC ASYMMETRIC MULTI-TASK LEARNING

## ABSTRACT

When performing multi-task predictions with time-series data, knowledge learned for one task at a specific time step may be useful in learning for another task at a later time step (e.g. prediction of sepsis may be useful for prediction of mortality for risk prediction at intensive care units). To capture such dynamically changing asymmetric relationships between tasks and long-range temporal dependencies in time-series data, we propose a novel temporal asymmetric multi-task learning model, which learns to combine features from other tasks at diverse timesteps for the prediction of each task. One crucial challenge here is deciding on the direction and the amount of knowledge transfer, since loss-based knowledge transfer (Lee et al., 2016; 2017) does not apply in our case where we do not have loss at each timestep. We propose to tackle this challenge by proposing a novel uncertainty-based probabilistic knowledge transfer mechanism, such that we perform knowledge transfer from more certain tasks with lower variance to uncertain ones with higher variance. We validate our Temporal Probabilistic Asymmetric Multi-task Learning (TP-AMTL) model on two clinical risk prediction tasks against recent deep learning models for time-series analysis, which our model significantly outperforms by successfully preventing negative transfer. Further qualitative analysis of our model by clinicians suggests that the learned knowledge transfer graphs are helpful in analyzing the model's predictions.

## 1 INTRODUCTION

Multi-task learning (MTL) (Caruana, 1997) is a method to train a model, or multiple models jointly for multiple tasks to obtain improved generalization, by sharing knowledge among them. One of the most critical problems in multi-task learning is the problem known as *negative transfer*, where unreliable knowledge from other tasks adversely affects the target task. To prevent negative transfer, researchers have sought ways to allow knowledge transfer only among closely related tasks, by either identifying the task groups or learning optimal sharing structure among task (Duong et al., 2015; Misra et al., 2016). However, not only the task relatedness but the relative reliability of the task-specific knowledge also matters, and recent asymmetric multi-task learning models Lee et al. (2016; 2017) tackle this challenge by allowing tasks with low loss to transfer more.

While the asymmetric knowledge transfer between tasks is useful, it does not fully exploit the asymmetry in the case of time-series analysis, which has an additional dimension of the time axis. With time-series data, knowledge transfer direction may need to be different depending on the timestep. For instance, suppose that we predict infection and mortality for patients in intensive care units based on their medical records. At earlier timesteps, prediction of infection may be more reliable than mortality, thus we may want knowledge transfer to happen from task infection to mortality; at later timesteps, we may want the opposite situation to happen. Moreover, knowledge transfer may happen across timesteps. For example, a high risk of infection in early timestep will alarm high risk of mortality at later timesteps. To exploit such temporal relationships between tasks, we need a model that does not perform static knowledge transfer between two tasks (Figure 1a), but dynamically changes the knowledge transfer amount and direction at each timestep, and also transfers knowledge across timesteps (Figure 1b). Toward this objective, we propose a multi-task learning framework for time-series data, where each task not only learns its own latent features at each timestep but also aggregates the latent features from the other tasks at the same or different timesteps via attention allocation.

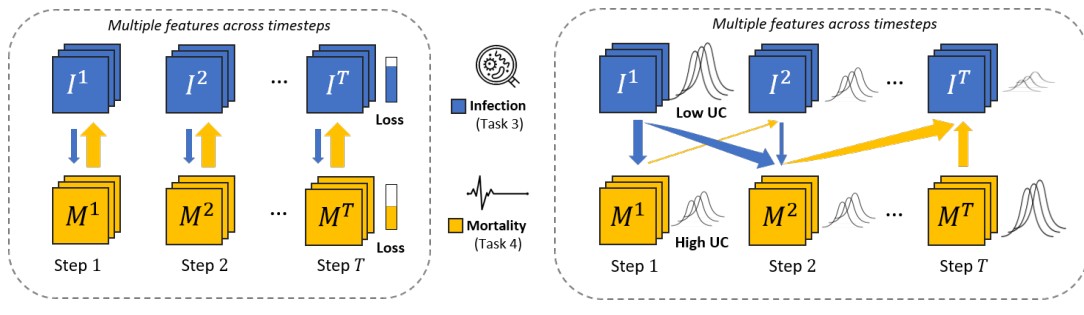

(a) AMTL (Lee et al., 2016)  (b) Temporal Probabilistic AMTL

Figure 1: **Concept:** **(a)** Existing asymmetric multi-task learning model utilize task loss to perform static knowledge transfer from one task to another; thus it cannot capture dynamically changing relationships between time-steps and tasks in the time-series domain. **(b)** Our model tackles the lack of loss by performing knowledge transfer based on the feature-level uncertainty. This allows dynamic asymmetric knowledge transfer among tasks and across timesteps.

Yet this brings in another challenge. On what basis should we promote asymmetric knowledge transfer? For asymmetric knowledge transfer between tasks, we could use task loss as a proxy of knowledge reliability. However, loss is not a direct measure of reliability, since a model trained with few instances may have a small loss, while the knowledge from the model could be highly biased and unreliable. Further, for prediction with time-series data, loss may not be available at every timestep. Thus, instead of task loss, we focus on the *uncertainty*, which can be obtained with probabilistic Bayesian models. Basically, if a latent feature learned at a certain timestep has large uncertainty (variance), we can consider its knowledge as unreliable. In such a case, the model may allocate small attention values for the feature; that is, the attention will be attenuated based on the uncertainty.

To validate the superiority of our model, we experiment with it on three clinical risk prediction datasets against multiple baselines. The results show that our model obtains significant improvements over strong multi-task learning baselines. Further, both the asymmetric knowledge transfer between tasks at two different timesteps as well as the uncertainty-based attenuation of attention weights are found to be useful in improving the generalization performance.

Our contribution in this work is threefold:

- We propose a novel asymmetric multi-task learning framework for time-series analysis, which utilizes feature-level uncertainty to perform knowledge transfer among tasks and across time-steps, thereby exploiting both the task-relatedness and temporal dependencies.
- We use a probabilistic Bayesian formulation for asymmetric knowledge transfer, where the amount of knowledge transfer depends on the uncertainty at the feature level.
- We validate our model on clinical risk prediction tasks, on which it achieves significant improvements over baselines and provides meaningful interpretations, including temporal relationships between tasks.

## 2 RELATED WORK

**Multi-task Learning** While the literature on multi-task learning (Caruana, 1997; Argyriou et al., 2008) is vast, we selectively mention the prior works that are closely related to ours. Historically, multi-task learning models have focused on *what to share* (Yang & Hospedales, 2016a;b; Ruder12 et al.), as the jointly learned models could share instances, parameters, or features (Kang et al., 2011; Kumar & Daume III, 2012; Maurer et al., 2013). With deep learning, multi-task learning can be implemented rather straightforwardly by making multiple tasks to share the same deep network. However, since solving different tasks will require heterogeneous knowledge, complete sharing of the underlying network may be suboptimal and brings in a problem known as *negative transfer*, where certain tasks are negatively affected by knowledge sharing. To prevent this, researchers are exploring more effective knowledge sharing structures. Duong et al. (2015) proposed a soft parameter sharing method that uses a regularizer to enforce the network parameters for each task to be similar, while Misra et al. (2016) proposed to learn the optimal combination of shared and task-specific representations in computer vision. Kendall et al. (2018) proposed a multi-task framework when the losses are weighed based on tasks' uncertainty, thereby reducing negative transfer from uncertain tasks. While finding a good sharing structure can alleviate negative transfer, negative transfer will still persist if we perform symmetric knowledge transfer among tasks. To resolve this symmetry issue, Lee et al. (2016) proposed an asymmetric MTL model with inter-task knowledge transfer

that allows task-specific parameters for tasks with smaller loss to transfer more. Lee et al. (2017) proposed a model for asymmetric task-to-feature transfer that allows reconstructing features with task-specific features while considering their loss, which is more suitable for deep neural networks and scalable. Our model is also targeting asymmetric multi-task learning, but is different from these previous works in that it utilizes *uncertainty* rather than loss as the measure of task reliability, and performs asymmetric knowledge transfer at each timestep, and across timesteps.

**Clinical time-series analysis** While our method is generic and applicable to any time-series prediction tasks, we mostly focus on clinical time-series analysis in this paper. Recently, there has been some progress on this topic, mostly focusing on interpretability and reliability of the model. Choi et al. (2016) proposed an attention-based model that generates attention for both the timesteps (hospital visits) and features (medical examination results), to provide interpretations of the predictions. However, attentions are often unreliable since they are learned in a weakly-supervised manner, and Heo et al. (2018) proposed to obtain reliable interpretation and prediction by proposing a probabilistic attention mechanism that considers uncertainty as to how to trust the input. Our work shares the motivation with these prior works as we target interpretability and reliability. Recently, Song et al. (2018) proposed a model that uses the Transformer architecture to perform time-series prediction with multi-task learning experiments. However, their model takes a straightforward approach where all tasks share a single base network, which is susceptible to negative transfer.

## 3 APPROACH

We now formally describe the problem setting and our probabilistic asymmetric multi-task learning framework for time-series prediction.

### 3.1 TIME-SERIES PREDICTION WITH FEATURE-LEVEL UNCERTAINTY

Our goal is to jointly train time-series prediction models for multiple tasks at once. Suppose that we are given training data for $D$ tasks, $\mathbb{D} = \{(\mathbf{X}_1, \mathbf{Y}_1), \ldots, (\mathbf{X}_D, \mathbf{Y}_D)\}$. Further suppose that each data instance $\mathbf{x}(\mathbf{x} \in \mathbf{X}_d$ for some $d)$ consists of $T$ timesteps. That is, $\mathbf{x} = (\mathbf{x}^{(1)}, \mathbf{x}^{(2)}, \ldots, \mathbf{x}^{(T)})$, where $\mathbf{x}^{(t)} \in \mathbb{R}^{1 \times m}$ denote the data instance for the timestep $t$. Additional, $y_d$ is the label for task $d$; for binary classification task, $y_d \in \{0, 1\}$, and for regression case, $y_d \in \mathbb{R}$. Given time-series data and tasks, we want to learn the task-specific latent features for each task and timestep, and then perform asymmetric knowledge transfer between them. Our multi-task learning framework is comprised of the following components:

**Shared Low-Level Layers (Figure 2a)** We allow our model to share low-level layers for all the tasks in order to learn a common data representation before learning task-specific features. At the lowest layer, we have a shared linear data embedding layer to embed the data instance for each timestep into a continuous shared feature space. Given a time-series data instance $\mathbf{x}$, we first linearly transform the data point for each timestep $t$, $\mathbf{x}^{(t)} \in \mathbb{R}^m$, which contains $m$ variables.

$$(\mathbf{v}^{(1)}, \mathbf{v}^{(2)}, ..., \mathbf{v}^{(T)}) = \mathbf{v} = \mathbf{x}\mathbf{W}_{emb} \in \mathbb{R}^{T \times k} \tag{1}$$

where $\mathbf{W}_{emb} \in \mathbb{R}^{m \times k}$ and $k$ is the number of hidden units. After embedding the data instances, we input them into a shared RNN layer for pre-processing:

$$\mathbf{r} = (\mathbf{r}^{(1)}, \mathbf{r}^{(2)}, ..., \mathbf{r}^{(T)}) = RNN(\mathbf{v}^{(1)}, \mathbf{v}^{(2)}, ..., \mathbf{v}^{(T)}) \tag{2}$$

**Task- and Timestep Embedding Layers** After embedding and pre-processing the input into a continuous space, we further encode them into task- and timestep-specific features. Since hard-sharing layers may result in negative transfer between tasks, we use separate embedding layers for each task to encode task-specific knowledge. For each task $d$, the separate network consists of $L$ layers (Figure 2b) of feed-forward networks, to learn disentangled knowledge for each timestep. The $L$ feed-forward layers for task embedding (Figure 2b) can be formalized as:

$$\mathbf{h}_d = \sigma((...\sigma(\sigma(\mathbf{r}\mathbf{W}_d^1 + \mathbf{b}_d^1)\mathbf{W}_d^2 + \mathbf{b}_d^2)...)\mathbf{W}_d^L + \mathbf{b}_d^L) \in \mathbb{R}^{T \times k} \tag{3}$$

where $\mathbf{W}_d^i \in \mathbb{R}^{k \times k}, \mathbf{b}_d^i \in \mathbb{R}^k$ and $\sigma$ is a non-linear activation function (e.g. leaky relu).

**Modeling feature-level uncertainty** While the above embedding can capture knowledge for each task and timestep, we want to further model their uncertainties as well, to measure the reliability of the knowledge captured. Towards this objective, we model the latent variables as probabilistic random variables, with two types of uncertainty (Kendall & Gal, 2017): 1) *epistemic uncertainty*, which

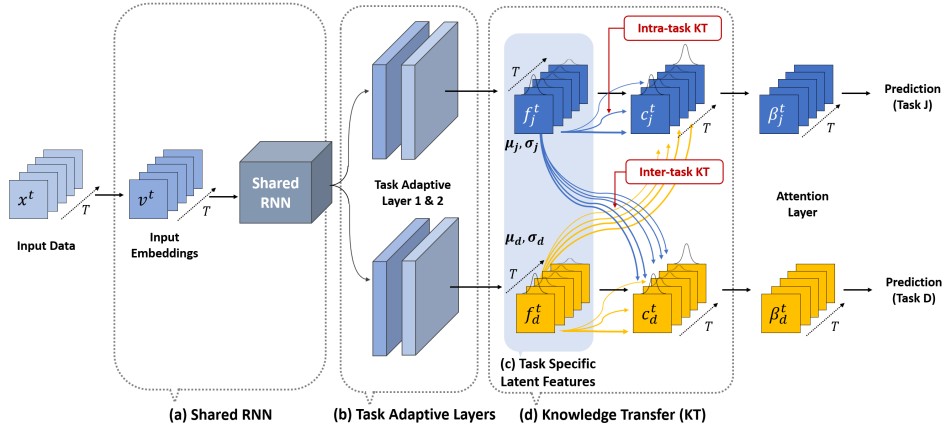

Figure 2: **Architecture overview.** The amount and form of (d) knowledge transfer is computed by networks $F$ and $G$, which will be described in detail on subsections 3.2 and 3.3.

comes from the model's unreliability from the lack of training data, and 2) *aleatoric uncertainty*, that comes from the inherent ambiguity in the data. We capture the former by using dropout variational inference (Gal & Ghahramani, 2016), and the latter by explicitly learning the model variance as a function of the input (Figure 2c).

Suppose that our generative model is: $\mathbf{z}_d \sim p_\theta(\mathbf{z}_d|\mathbf{x}, \boldsymbol{\omega}), y_d \sim p_\theta(y_d|\mathbf{x}, \mathbf{z}_d, \boldsymbol{\omega})$ where $\boldsymbol{\omega}$ is the set of all parameters, $p_\theta(\mathbf{z}_d|\mathbf{x}, \boldsymbol{\omega})$ and $p_\theta(y_d|\mathbf{x}, \mathbf{z}_d, \boldsymbol{\omega})$ are networks parameterized by $\theta$. We denote all joint, marginal and conditional distributions arising from this generative model by $p$.

$$\mathbf{z}_d|\mathbf{x}, \boldsymbol{\omega} \sim \mathcal{N}\left(\mathbf{z}_d; \boldsymbol{\mu}_d, diag(\boldsymbol{\sigma}_d^2)\right) \tag{4}$$

$$\boldsymbol{\mu}_d = \sigma(\mathbf{h}_d\mathbf{W}_d^\mu + \mathbf{b}_d^\mu) \tag{5}$$

$$\boldsymbol{\sigma}_d = softplus(\mathbf{h}_d\mathbf{W}_d^\sigma + \mathbf{b}_d^\sigma) \tag{6}$$

The posterior distributions $p(\mathbf{z}_d|\mathbf{x}, y_d, \boldsymbol{\omega})$ and $p(\boldsymbol{\omega}|\mathbb{D})$ are intractable so we use variational inference to approximate the joint distribution. The variational distribution is defined by:

$$q(\mathbf{z}_d, \boldsymbol{\omega}|\mathbf{x}, y_d, \mathbb{D}) = q_M(\boldsymbol{\omega}|\mathbb{D})q(\mathbf{z}_d|\mathbf{x}, y_d, \boldsymbol{\omega}) \tag{7}$$

We set $q_M(\boldsymbol{\omega}|\mathbb{D})$ by dropout approximation (Gal & Ghahramani, 2016) with parameter $M$ and simply set $q(\mathbf{z}_d|\mathbf{x}, y_d, \boldsymbol{\omega})$ to the prior network $p_\theta(\mathbf{z}_d|\mathbf{x}, \boldsymbol{\omega})$ (which works well in practice (Sohn et al., 2015; Heo et al., 2018)) for the model consistency at both training and test time.

### 3.2 UNCERTAINTY-AWARE KNOWLEDGE TRANSFER

We perform asymmetric knowledge transfer in the feature space. Suppose that we have latent feature vectors $\mathbf{f}$ and $\mathbf{g}$ for two tasks at different timesteps. Then the model needs to decide on both 1) the amount of knowledge to transfer, and 2) 'the transferred knowledge:

**1) The amount of knowledge to transfer** Existing AMTL models (Lee et al., 2016; 2017) often use task loss to decide on the amount of knowledge transfer, such that task-specific parameters with low task loss to transfer more, while tasks with high loss only receive knowledge transfer. Yet, the task loss may be unreliable as a measure of the knowledge from the task and may not be available at each timestep. To overcome these limitations, we propose to learn the amount of knowledge transfer based on the feature-level uncertainty. With our probabilistic time-series prediction model described in the previous subsection, the source features $\mathbf{f}$ and the target features $\mathbf{g}$ follow some distributions. Specifically, $\mathbf{f} \sim p(\mathbf{f}; \boldsymbol{\mu}_f, \boldsymbol{\sigma}_f)$ and $\mathbf{g} \sim p(\mathbf{g}; \boldsymbol{\mu}_g, \boldsymbol{\sigma}_g)$. Our model learns the distribution of the transfer weight $\alpha$ from $\mathbf{f}$ to $\mathbf{g}$ by a small network $F_\theta$ parameterized by $\theta$, which takes both (instances of) $\mathbf{f}$ and $\mathbf{g}$ as its input:

$$\alpha = \sigma(a) \tag{8}$$

$$a \sim p_\theta(a|\mathbf{f}, \mathbf{g}) = \mathcal{N}(a; \mu_a, \sigma_a^2) \tag{9}$$

$$(\mu_a, \sigma_a) = F_\theta(\mathbf{f}, \mathbf{g}) \tag{10}$$

In general, $F_\theta$ is a function from $\mathbb{R}^{2k}$ to $\mathbb{R}^2$. In practice, to avoid the concatenation of $\mathbf{f}$ and $\mathbf{g}$ (which can be slow), we adopt a simple implementation: $\mu_a = \sigma(\sigma(\boldsymbol{f}\boldsymbol{W}_f + \boldsymbol{b}_f) \cdot \sigma(\boldsymbol{g}\boldsymbol{W}_g + \boldsymbol{b}_g))$ and $\sigma_a = soft\_plus(\mu_a w + b)$, where $\sigma$ is some activation function (which should not be confused with $\sigma_a$) and $\cdot$ is the inner product.

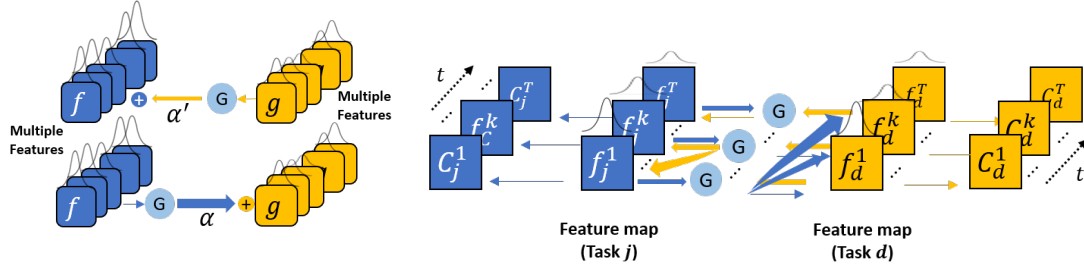

(a) Uncertainty-aware knowledge transfer      (b) Temporal asymmetric knowledge transfer

Figure 3: **Temporal probabilistic asymmetric knowledge transfer**. 3a illustrates the uncertainty-aware knowledge transfer between two latent features. In this case, $f$ is more reliable than $g$, so the model will learn to transfer more from $f$ to $g$ and transfer less from $g$ to $f$. 3b shows how we apply it for knowledge transfer between two tasks at the same timestep and at different timesteps.

By using Stochastic Gradient Variational Bayes (SGVB), the network $F$ can learn a meaningful distribution of the transfer weight, such that it sets the value of $\alpha$ high when the $\mathbf{f}$ and $\mathbf{g}$ are related, with low uncertainty on $\mathbf{f}$ and high uncertainty on $\mathbf{g}$.

**2) The form of transferred knowledge** Because $\mathbf{f}$ and $\mathbf{g}$ may have completely different representations, directly combining the two by adding $\alpha\mathbf{f}$ to $\mathbf{g}$ is suboptimal. Instead, we use a network $G_\phi$ parameterized by $\phi$ to non-linearly transform $\mathbf{f}$ into a compatible representation for $\mathbf{g}$ (Figure 3a). The combined features $\mathbf{C}$ then can be expressed as follows:

$$\mathbf{C} = \mathbf{g} + \alpha G_\phi(\mathbf{f}) \tag{11}$$

$$G_\phi(f) = \sigma(\mathbf{f}\mathbf{W} + \mathbf{b}) \tag{12}$$

where $\mathbf{W} \in \mathbb{R}^{k \times k}, \mathbf{b} \in \mathbb{R}^k$ and $\sigma$ is a non-linear activation (e.g. leaky relu). Furthermore, when transferring knowledge from a source task to a target task, the target task should only borrow knowledge from the source task without affecting its feature map via backpropagation. To this end, we block the gradient of $\mathbf{f}$ in the formulation of $\mathbf{C}$ in 11. We also empirically found that this gradient blocking makes the obtained knowledge transfer graph significantly more interpretable.

### 3.3 ASYMMETRIC KNOWLEDGE TRANSFER ACROSS TASKS AND TIME STEPS

Now we apply the proposed probabilistic asymmetric knowledge transfer method to perform knowledge transfer across timesteps, both within each task and across tasks, to exploit intra- and inter-task temporal dependencies. In order to transfer knowledge from task $j$ to task $d$ with temporal dependencies, we allow the latent features of task $d$ at time step $t$ ($\mathbf{f}_d^{(t)}$, with $\mathbf{z}_d = (\mathbf{f}_d^{(1)}, \mathbf{f}_d^{(2)}, ..., \mathbf{f}_d^{(T)})$) to obtain knowledge from task $j$ at all previous time steps (see Figure 3b), and then combine them into a single feature map $\mathbf{C}_d^{(1)}, \mathbf{C}_d^{(2)}, ..., \mathbf{C}_d^{(T)}$:

$$\mathbf{C}_d^{(t)} = \mathbf{f}_d^{(t)} + \sum_{j=1}^{D}\sum_{i=1}^{t} \alpha_{j,d}^{(i,t)} * G_{j,d}\left(\mathbf{f}_j^{(i)}\right) \forall t \in \{1, 2, ..., T\}$$

Here, we chose to constrain the knowledge transfer to happen only from past to future timesteps because of the time complexity at inference time. With our proposed model, for **each update** at the clinical environment in a online manner, we only need to transfer the knowledge from previous time steps to the current one, making the complexity to be $\mathbf{O(T)}$. This is on a par with other models like RETAIN (Choi et al., 2016) or UA (Heo et al., 2018), making it highly scalable. However, if we allow the knowledge to transfer from future timestep to past timestep, we also need to update the knowledge at previous timesteps for a single update. The time complexity of the model in this case is $O(T^2)$, which is undesirable. In the ablation study in Section 4.5, we show that this constraint also brings in small performance gain. The total complexity of the **whole** training or inference is still $\mathbf{O(T^2)}$. However, it is expected due to the inter-timestep transfer, and is on a par with state-of-the-art models such as Transformer(Vaswani et al., 2017) or RETAIN(Choi et al., 2016).

Finally, we use the combined features $\mathbf{C}_d^{(1)}, \mathbf{C}_d^{(2)}, ..., \mathbf{C}_d^{(T)}$, which contain temporal dependencies among tasks, for prediction for each task $d$. We use an attention mechanism:

$$\boldsymbol{\beta}_d^{(t)} = tanh\left(\mathbf{C}_d^{(t)}\mathbf{W}_d^\beta + \mathbf{b}_d^\beta\right) \quad \forall t \in \{1, 2, ..., T\}$$

where $\mathbf{W}_d^\beta \in \mathbb{R}^{k \times k}$ and $\mathbf{b}_d^\beta \in \mathbb{R}^k$, and $\alpha_{j,d}^{(i,t)} (i \le t)$ is the knowledge transfer weight from task $j$ at timestep $i$ to task $d$ at timestep $t$. Then the model can perform prediction as follows,

$$p(\widehat{y_d}|\mathbf{x}) = Sigmoid\left(\frac{1}{T}(\boldsymbol{\beta}_d^{(1)} \odot \mathbf{v}^{(1)} + \boldsymbol{\beta}_d^{(2)} \odot \mathbf{v}^{(2)} + ... + \boldsymbol{\beta}_d^{(T)} \odot \mathbf{v}^{(T)})\mathbf{W}_d^o + b_d^o\right) \qquad (13)$$

for classification tasks, where $\odot$ denotes the element-wise multiplication between attention ${\beta_d}^{(t)}$ and shared input embedding $\mathbf{v}^{(t)}$ (from Eq. 1), $\mathbf{W}_d^o \in \mathbb{R}^{k \times 1}$ and $b_d^o \in \mathbb{R}^1$. Predictions for other tasks are done similarly.

It should be noted that our model does not require each instance to have the labels for every tasks. We can just maximize the likelihood $p(y_d|\mathbf{x})$ whenever the label $y_d$ is available for input $x$ for task $d$. Furthermore, it is not required that all the instances to have the same number of time-steps $T$. But in practice, we can use zero-padding to make the training easier.

## 4 EXPERIMENTS

We validate our model on multiple clinical risk prediction tasks against relevant baselines.

### 4.1 TASKS AND DATASETS

We experiment on two clinical time-series datasets. For all datasets, we randomly split the data into training, validation, and test set. For more details on the datasets and tasks, including explanations for dataset *MIMIC III - Infection*, the base network configurations and hyperparameters, and more experimental results on an additional dataset (MIMIC III - Heart Failure), please **see the appendix**.

**1) MIMIC III - Infection (Figure 6).** We compiled a dataset out of the MIMIC III dataset (Johnson et al., 2016), which contains electronic health records (EHR) of 53,423 distinct hospital admissions between 2001 and 2012 to the intensive care unit (ICU) of a hospital. We use records of patients over age 15, from the first 48 hours after the admission, in 48 timesteps. Following clinician's guidelines, we select 12 infection-related variables for the features at each timestep, including *heart rate*, *arterial blood pressure*, and *Glasgow Coma Scale(GCS)*. [1]. Tasks we consider for this dataset are the ones that are necessary for the diagnostic process of patient's infectious status; The first task, *body temperature elevation* (Task 1) provides the signs of infection, and *Infection* (Task 2) as the confirmation of infection by the result of microbiology tests. We also added *Mortality* (Task 3) as a possible outcome of infection. After pre-processing the data, we were able to select approximately 2000 data points (that have sufficient amount of features). We use a random split of approximately 1000/500/500 for training/validation/test.

**2) PhysioNet.** This dataset (Citi & Barbieri, 2012) contains 4,000 medical records from ICU. Each record contains 48 hours of records, each of which contains 31 physiological signs including *heart rate*, *respiratory rate*, *temperature*, etc. Task used in the experiment includes four binary classification tasks, namely, 1) *Mortality prediction*, 2) *Length-of-stay less than 3 days*: whether the patient would stay in ICU for less than three days, 3) *Cardiac Surgery ICU(Cardiac):* whether the patient is recoveing from cardiac surgery, and 4) *Recovery in Surgical ICU(Recovery):* whether the patient is staying in Surgical ICU to recover from surgery. We use a random split of 2800/400/800 for training/validation/test.

### 4.2 BASELINES

Now we describe the baseline single-task and multi-task models for time-series prediction, along with our models.

**1) STL-LSTM.** The base single-task Long Short-Term Memory Network for time-series prediction.
**2) RETAIN.** The attentional RNN proposed in (Choi et al., 2016) which focuses on interpretability, for clinical prediction with electronic health records.
**3) UA.** The uncertainty-aware probabilistic attention model proposed in (Heo et al., 2018) for interpretable time-series prediction.
**4) MTL-LSTM.** The naive hard-sharing multi-task learning method where all tasks share the same network except for the separate output layers for prediction, with LSTM as the base network.
**5) MTL-Transformer.** The same as MTL-LSTM, but with Transformer (Vaswani et al., 2017) as

---

[1]The full detail of the variables can be found in the supplementary file

Table 1: MIMIC-III Infection

| | Models | Tasks | | | |
|---|---|---|---|---|---|
| | | Fever | Infection | Mortality | Average |
| STL | LSTM | 0.6738± 0.02 | 0.6860± 0.02 | 0.6373± 0.02 | 0.6657± 0.02 |
| | RETAIN (Choi et al., 2016) | 0.6826± 0.01 | 0.6655± 0.01 | 0.6054± 0.02 | 0.6511± 0.01 |
| | UA (Heo et al., 2018) | 0.6987± 0.02 | 0.6504± 0.02 | 0.6168± 0.05 | 0.6553±0.02 |
| MTL | LSTM | 0.7006± 0.03 | 0.6686± 0.02 | 0.6261± 0.03 | 0.6651 ±0.02 |
| | TRANS (Vaswani et al., 2017) | 0.7025± 0.01 | 0.6479± 0.02 | 0.6420± 0.02 | 0.6641±0.02 |
| | RETAIN (Choi et al., 2016) | 0.7059± 0.02 | 0.6635± 0.01 | 0.6198± 0.05 | 0.6630± 0.02 |
| | UA (Heo et al., 2018) | **0.7124± 0.01** | 0.6489± 0.02 | 0.6325± 0.04 | 0.6646±0.02 |
| | RETAIN-Kendall (Kendall et al., 2018) | 0.6938± 0.01 | 0.6182± 0.03 | 0.5974± 0.02 | 0.6364± 0.02 |
| | AMTL-LSTM (Lee et al., 2016) | 0.6858± 0.01 | 0.6773± 0.01 | 0.6765± 0.01 | 0.6798±0.01 |
| | TP-AMTL (our model) | **0.7156± 0.01** | **0.7131± 0.01** | **0.7098± 0.03** | **0.7128±0.01** |

Table 2: PhysioNet

| | Models | Tasks | | | | |
|---|---|---|---|---|---|---|
| | | Stay < 3 | Cardiac | Recovery | Mortality | Average |
| STL | LSTM | 0.7673± 0.09 | 0.9293± 0.01 | 0.8587± 0.01 | 0.7100± 0.01 | 0.8163± 0.03 |
| | RETAIN (Choi et al., 2016) | 0.7407± 0.04 | 0.9236± 0.01 | 0.8148± 0.04 | 0.7080± 0.02 | 0.7968± 0.03 |
| | UA (Heo et al., 2018) | 0.8556± 0.02 | 0.9335± 0.01 | 0.8712± 0.01 | 0.7283± 0.01 | 0.8471± 0.01 |
| MTL | LSTM | 0.7418± 0.09 | 0.9233± 0.01 | 0.8472± 0.02 | 0.7228± 0.01 | 0.8088± 0.03 |
| | TRANS (Vaswani et al., 2017) | 0.8532± 0.03 | 0.9291± 0.01 | 0.8770± 0.01 | 0.7358± 0.01 | 0.8488± 0.01 |
| | RETAIN (Choi et al., 2016) | 0.7613± 0.03 | 0.9064± 0.01 | 0.8160± 0.04 | 0.6944± 0.03 | 0.7945± 0.03 |
| | UA (Heo et al., 2018) | 0.8573± 0.03 | 0.9348± 0.01 | 0.8860± 0.01 | **0.7569± 0.02** | 0.8587± 0.02 |
| | RETAIN-Kendall (Kendall et al., 2018) | 0.7418± 0.02 | 0.9219± 0.02 | 0.7883± 0.03 | 0.6787± 0.02 | 0.7827 ± 0.02 |
| | AMTL-LSTM (Lee et al., 2016) | 0.7600± 0.08 | 0.9254± 0.01 | 0.8066± 0.01 | 0.7167± 0.01 | 0.8022± 0.03 |
| | TP-AMTL (our model) | **0.9012± 0.01** | **0.9368± 0.01** | **0.8923± 0.01** | **0.7571± 0.01** | **0.8719± 0.01** |

the base network.
**6) MTL-RETAIN.** The same as MTL-LSTM, but with RETAIN as the base network.
**7) MTL-UA.** The same as MTL-LSTM, but with UA (Heo et al., 2018) as the base network.
**8) AMTL-LSTM.** Asymmetric multi-task learning (Lee et al., 2016) adopted for our time-series prediction framework, where we learn the knowledge transfer graph between task-specific parameters, for asymmetric knowledge transfer based on the task loss. Since the parameters for each task are shared across all timesteps, this will result in static asymmetric transfer between tasks.
**9) MTL-RETAIN-Kendall.** This is the uncertainty-based loss-weighing scheme proposed in Kendall et al. (2018) with MTL-RETAIN as the base MTL model.
**10) TP-AMTL.** Our probabilistic temporal asymmetric multi-task learning model that performs both intra-task and inter-task knowledge transfer.
For more details about the baselines, please refer to the **appendix**.

### 4.3 QUANTITATIVE EVALUATION ON CLINICAL TIME-SERIES PREDICTION TASKS

We first evaluate the prediction accuracies of the baseline STL and MTL models and ours on the two clinical time-series datasets, by measuring Area Under the ROC curve (AUROC). Table 1,2 show the result for the MIMIC-III Infection dataset and Physionet dataset respectively. We observe that hard-sharing MTL models outperform STL on some tasks, but suffers from performance degeneration on other tasks, which could be the effect of negative transfer. MTL models especially work poorly on MIMIC-III Infection, which has clear temporal relationships between tasks. Probabilistic models (e.g. UA) generally outperform their deterministic counterparts (e.g. RETAIN). However, AMTL-RETAIN-Kendall, which learns the weight for each task loss based on uncertainty, significantly underperforms even the STL-LSTM, which may be due to the fact that losses in our settings are at almost similar scale unlike with the task losses in Kendall et al. (2018) that have largely different scales. AMTL-LSTM improves on some tasks, but degenerates the performance on the others, which we attribute to the fact that it does not consider inter-timestep transfer. On the other hand, our model, TP-AMTL, obtains significant improvements over all STL and MTL baselines on both datasets. It also does not show performance degeneration on any of the tasks, suggesting that it has successfully dealt away with negative transfer in multi-task learning with time-series prediction models.

For further analysis of the relationships between uncertainty and knowledge transfer, we visualize knowledge transfer from multiple sources (Figure 4a) normalized over the number of tagets, and to multiple targets (Figure 4b) normalized over the number of sources, along with their uncertainties.

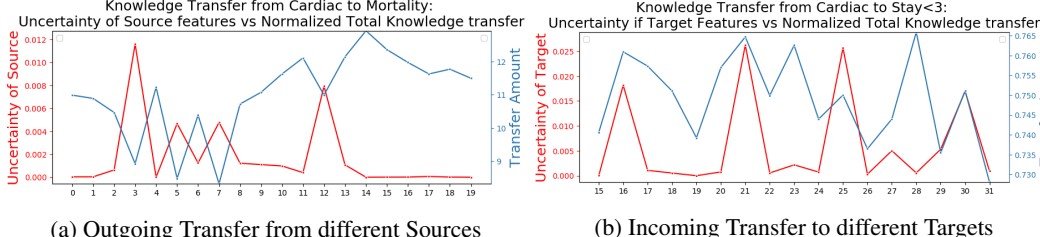

(a) Outgoing Transfer from different Sources
(b) Incoming Transfer to different Targets

Figure 4: Examples showing the relationship between the amount of knowledge transfer and the uncertainty of source and target features. (a) The sources with low uncertainty transfer more knowledge. (b) The targets with high uncertainty receive more knowledge.

Specifically, the uncertainty of a task at a certain timestep is represented by the average of the variance of all feature distributions. The normalized amount of knowledge transfer from task $j$ at time step $t$ to task $d$ is computed as $(\alpha_{j,d}^{(t,t)}+\alpha_{j,d}^{(t,t+1)}+...+\alpha_{j,d}^{(t,T)})/(T-t+1)$. Similarly, the normalized amount of knowledge transfer to task $d$ at time step $t$ from task $j$ is $(\alpha_{j,d}^{(1,t)}+\alpha_{j,d}^{(2,t)}+...+\alpha_{j,d}^{(t,t)})/t$

We observe that source features with low uncertainties transfer knowledge more, while at the target, features with high uncertainties receive more knowledge transfer. However, note that they are not perfectly correlated, since the amount of knowledge transfer is also affected by the pairwise similarities between the source and the target features as well.

### 4.4 INTERPRETATION OF THE LEARNED KNOWLEDGE GRAPHS

With the help of a physician, we further analyze how generated transfer weights and uncertainties are related with the patient's actual medical conditions and how we can interpret temporal relationships between tasks with our model (see Table 6 and Figure 5). We first consider an example record of a patient from the MIMIC-III Infection dataset who was suspected of infection on admission having fever as an initial symptom, which was diagnosed to be caused by bacterial infection later. Figure 5a shows the amount of knowledge transfer from task *fever* at 3:00 to all later timesteps of task *infection*. At this timestep, the patient's condition changes significantly. In Table 6, we see that the patient had a fever and the white blood cell count increased to the state of leukocytosis, and both the systolic and diastolic blood pressure decrease over time. Most importantly, the patient is diagnosed to have an infection, as the bacterial culture test result turns out to be positive at 2:57. With the drop of uncertainty of the task *infection* around the time window where the event happens, the amount of knowledge transfer from *fever* to *infection* drops as well, as the knowledge from the source task becomes less useful.

Table 3: Clinical Events in selected medical records for case studies. **MechVent** - Mechanical Ventilation, **FiO2** - Fractional inspired Oxygen, **SBP** - Systolic arterial blood pressure, **DBP** - Diastolic arterial blood pressure, **HR** - Heart Rate, **Temp** - Body Temperature, **Urine** - Urine output, **GCS** - Glasgow Coma Score, **WBC** - White Blood Cell Count, **Culture** - Culture Results.

| | SBP | DBP | Temp | WBC | Culture Results | | Mech Vent | FiO2 | SBP | DBP | MAP | HR | Temp | Urine | GCS |
|---|---|---|---|---|---|---|---|---|---|---|---|---|---|---|---|
| 23:00 | 138 | 64 | 38.4 | N/A | N/A | 23' 31" | 0 | N/A | 115 | 64 | 84 | 77 | 37.7 | 100 | 15 |
| 1:00 | 100 | 53 | 40.1 | 12500 | N/A | 29' 31" | 1 | 0.7 | 106 | 55 | 70 | 74 | N/A | 5 | 6 |
| **2:57** | **89** | **46** | **N/A** | **N/A** | **(+)Klebsiella Pneumoniae** | 30' 31" | **1** | **0.6** | **109** | **57** | **73** | **75** | **39.1** | **6** | **7** |

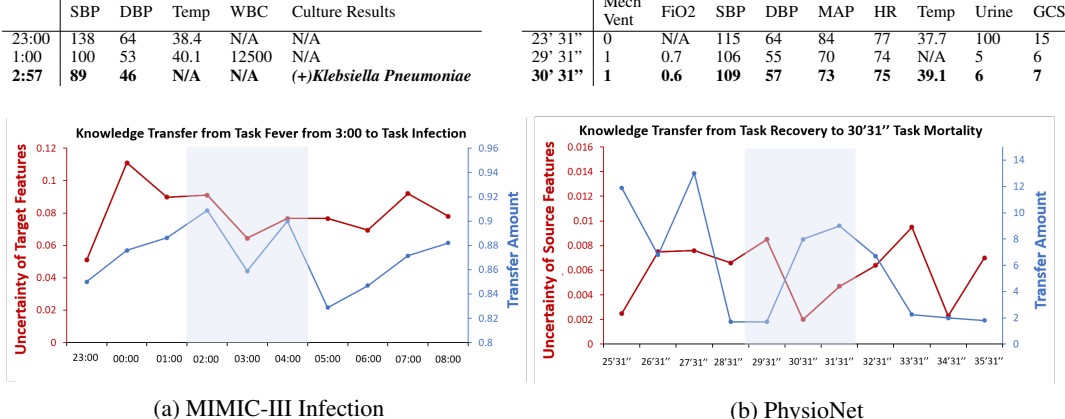

(a) MIMIC-III Infection
(b) PhysioNet

Figure 5: **Visualizations of the amount of uncertainty and normalized knowledge transfer for example cases** where the changes in the amount of uncertainty at certain timesteps are correlated with clinical events. We denote the timesteps with noticeable changes in uncertainty and knowledge transfer with blue boxes.

For another case study, we considered a record of a patient from PhyisoNet dataset who recovered from surgery and passed away during admission (See Figure 5b) for mortaility prediction at $30'31''$. We observe that at this timestep, as the uncertainty of the source task drops, the amount of knowledge transfer increases, since the knowledge from the source task becomes more reliable. From Table 6, we observe that at this time step, the patient is applied mechanical mechanical ventilation and the GCS score declined from 15 to 7, and the body temperature increased from 37.7 to 39.9. This patient is recovering from surgery, and the changes in features that are vital to survival means that general condition of this patient has altered, needing physician's attention. Based on this change in the patient's condition, the model may have predicted that the patient is recovering from surgery, which explains why the uncertainty on *recovery* decreased, and knowledge transfer to *mortality* increases to transfer this reliable knowledge.

For more example cases, please **see the appendix**. These interpretations suggest that by analyzing the learned knowledge graph using our model, we could identify timesteps where interesting interactions occur between tasks. This interpretability may become even more useful in large-scale settings where both the number of time-series data instances and the number of timestep is extremely large, on which manual analysis becomes impractical.

### 4.5  ABLATION STUDY

**Effectiveness of the inter-task and inter-timestep knowledge transfer.**   To show the effectiveness of the inter-task and inter-timestep knowledge transfer, we further compare our model on the PhysioNet dataset against several variations of **our model** which we describe below:
**1) AMTL-intratask:** The probabilistic AMTL model with uncertainty-aware knowledge transfer, but performs knowledge transfer only within the same task at the transfer layer. Note that, however, this model can still share inter-task knowledge in a symmetrical manner since it stilll has shared lower layers (the embedding and the LSTM layers).
**2) AMTL-samestep:** The probabilistic model with uncertainty-aware knowledge transfer, which performs knowledge transfer only between the features at the same timestep, at the transfer layer. Again, note that this model can still capture the temporal dependencies among the timesteps to certain degree, as it has shared lower layers.
**3) TD-AMTL:** The deterministic version of the temporal asymmetric multi-task learning model that does not make use of feature-level uncertainty when performing knowledge transfer.

Table 4: Ablation Study - Physionet

| Model | Tasks | | | | |
|---|---|---|---|---|---|
| | Stay<3 | Cardiac | Recovery | Mortality | Average |
| AMTL-intratask | $0.8829 \pm 0.01$ | $\mathbf{0.9338 \pm 0.01}$ | $0.8812 \pm 0.01$ | $0.7521 \pm 0.01$ | 0.8625 |
| AMTL-samestep | $0.8669 \pm 0.01$ | $0.9273 \pm 0.01$ | $\mathbf{0.8902 \pm 0.01}$ | $0.7382 \pm 0.01$ | 0.8557 |
| TD-AMTL | $0.7381 \pm 0.06$ | $0.9155 \pm 0.01$ | $0.8629 \pm 0.01$ | $0.7365 \pm 0.01$ | 0.8133 |
| TP-AMTL (unconstrained) | $\mathbf{0.8999 \pm 0.01}$ | $0.9186 \pm 0.01$ | $\mathbf{0.8892 \pm 0.01}$ | $\mathbf{0.7610 \pm 0.01}$ | **0.8672** |
| TP-AMTL (epistemic) | $\mathbf{0.8940 \pm 0.01}$ | $\mathbf{0.9358 \pm 0.01}$ | $\mathbf{0.8920 \pm 0.01}$ | $0.7557 \pm 0.01$ | **0.8694** |
| TP-AMTL (aleatoric) | $0.7939 \pm 0.03$ | $0.9176 \pm 0.01$ | $0.8529 \pm 0.02$ | $0.7372 \pm 0.03$ | 0.8254 |
| TP-AMTL (full model) | $\mathbf{0.9012 \pm 0.01}$ | $\mathbf{0.9368 \pm 0.01}$ | $\mathbf{0.8923 \pm 0.01}$ | $\mathbf{0.7571 \pm 0.01}$ | **0.8719** |

Table 4 shows that our model outperforms the "intratask" and "samestep" variants, which demonstrates the effectiveness of inter-task and inter-step knowledge transfer. Moreover, the deterministic counterpart largely underperforms any variants, which may be due to overfitting of the knowledge transfer graph, that can be effectively prevented by our Bayesian framework.

**Future-to-past transfer.**   We also compare our model against a variation of our method with no temporal constraint on the inter-step knowledge transfer (TP-AMTL (unconstrained)), such that the knowledge transfer can happen from the later timestep to earlier ones. Table 4 shows that the constrained model outperforms the unconstrained model, while having lower time complexity for each update ($O(T)$) than the unconstrained model $O(T^2)$.

**Two kinds of uncertainty.**   Furthermore, we examine the effect of two kinds of uncertainty with two variants of the model: **TP-AMTL (epistemic)** uses only MC-dropout to model epistemic uncertainty and $p_\theta(\mathbf{z}_d|\mathbf{x}, \boldsymbol{\omega})$ is simplified into $\mathcal{N}(\mathbf{z}_d; \boldsymbol{\mu}_d, \mathbf{0})$ (i.e. its pdf becomes the dirac delta function at $\boldsymbol{\mu}_d$ and $\mathbf{z}_d$ is always $\boldsymbol{\mu}_d$); **TP-AMTL (aleatoric)** uses only $p_\theta(\mathbf{z}_d|\mathbf{x}, \boldsymbol{\omega})$ to model the aleatoric uncertainty, withouth MC-dropout.

Table 4 shows that, for this dataset, epistemic uncertainty attributes more to the performance gain. However, it should be noted that the impacts of two kinds of uncertainty vary from dataset to dataset. By modelling both kinds of uncertainty, the model is guaranteed to get the best performance.

## 5 CONCLUSION

We proposed a novel probabilistic asymmetric multi-task learning framework that allows asymmetric knowledge transfer between tasks at different timesteps, based on the uncertainty. While existing asymmetric multi-task learning methods consider asymmetric relationships between tasks as fixed, with time-series data, the task relationship may change at different timesteps. Moreover, knowledge obtained for a task at a specific timestep could be useful for other tasks in later timesteps. Thus, to model varying direction of knowledge transfer and across-timestep knowledge transfer, we proposed a novel probabilistic multi-task learning framework that performs knowledge transfer based on the uncertainty of the latent representations for each task and timestep. We validated our model on clinical time-series prediction tasks on two datasets, on which our model significantly outperforms the baseline symmetric and asymmetric multi-task learning models. We further studied the learned knowledge graphs to show that our model can be used to provide useful interpretations on how the model made certain predictions, which is crucial in building a safe AI system.

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

## A DETAILED DESCRIPTION OF DATASETS AND EXPERIMENTAL SETUP

### A.1 FEATURES AND TASKS

**1) MIMIC III-Infection** Dataset used in this experiment was custom-constucted from whole MIMIC III dataset to represent temporal dependencies between tasks. We use the first 48 hours after admission for each patients to only consider the patient condition on admission, as infection occurring after 48 hour is more likely to be acquired at the ICUs. However, our method can be further applied to time-series prediction tasks with longer time steps. Following clinician's guidelines, we select 15 infection-related variables including *Heart rate*, *Systolic Blood Pressure*, *Diastolic Blood Pressure*, *Glasgow Coma scale(GCS)*, *invasive procedures:* this includes endoscopic procedure, intubation, dialysis, chest tube placement, Lumbar drainage, and biopsy etc., serum *albumin* and *total protein* which represents nutritional status of patients, and *intravenous steroids*, for the features at each timestep (see Table 7).

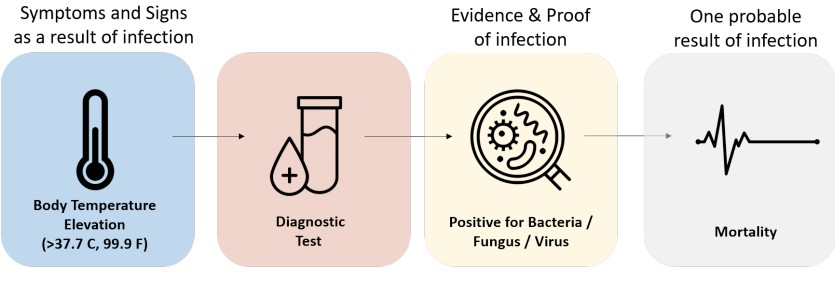

Figure 6: **Task overview.** Tasks used in the experiment included the diagnostic process of patient's infectious status (see Figure 6). When a patient in ICU is infected to any of the pathogen, body temperature elevates (*Fever (Task 1)*) as a sign of infection. Next, when physician prescribe diagnostic test such as blood culture, *Infection (Task 2)* is confirmed when the culture result turns out to be positive with bacteria, fungus, or virus. Lastly, *Mortality (Task 3)* can be resulted from infection.

**2) PhysioNet** We select 29 physiological signs as listed in this section : *age, gender, height, weight, Systolic Blood Pressure, Diastolic Blood Pressure, mean arterial pressure, heart rate, respiratory rate, body temperature, glucose, bilirubin, serum electrolytes (sodium, potassium, magnesium, bicarbonate), lactate, pH, Hematocrit, platelets, Partial Pressure of Oxygen($PaO_2$), Partial Pressure of carbon dioxide($PaCO_2$), Oxygen Saturation($SaO_2$), Fraction of inspired Oxygen($FiO_2$), Glasgow Coma scale(GCS), blood urea nitrogen(BUN), Creatinine, Urine, mechanical ventilation status*

**3) MIMIC III - Heart Failure** To test the generalizability of our model, we compiled this additional task set out of the MIMIC III dataset (Johnson et al., 2016). We were able to collect $3,577$ of distinct data instances where adult(age between 18 to 100) patients have admitted to the intensive care unit (ICU) of a hospital. This dataset contains 15 features which are associated to the risk of heart failure occurrence, including *Heart rate(HR)*, *Systolic Blood Pressure(SBP)*, *Diastolic Blood Pressure(DBP)*, *Body Temperature(BT)*, *Fraction of inspired oxygen (FiO$_2$)*, *Mixed venous oxygen saturation ($M_vO_2$)*, *Oxygen Saturation of arterial blood ($S_aO_2$)*, *Brain natriuretic peptide(BNP)*, *Ejection Fraction (EF)*,*Glasgow Coma scale(GCS) - Verbal, Motor, Eye*. Tasks we consider for this dataset are the ones that might lead to the development of heart failure. Total 4 tasks selected, and the first task, *Ischemic* (Task 1) is the patient condition where a patient is diagnosed with ischemic heart disease. The second task *Valvular* (Task 2) is related to the diagnosis of valvular heart disease, and the third task *Heart Failure* contains the condition where a patient is diagnosed with various types of heart failure. Lastly, *Mortality* (Task 4) can be a possible outcome of heart failure. We use a random split of approximately 1850/925/925 for training/validation/test.

### A.2 BASELINES

Here we describe the baselines with more details.

**1) STL-LSTM.** The single-task learning method which uses RNNs to capture the temporal dependencies.

$$(\mathbf{v}_d^{(1)}, \mathbf{v}_d^{(2)}, ..., \mathbf{v}_d^{(T)}) = \mathbf{v}_d = \mathbf{x}\mathbf{W}_{emb}^d \in \mathbb{R}^{T \times k} \tag{14}$$

$$(\mathbf{h}_d^{(1)}, \mathbf{h}_d^{(2)}, ..., \mathbf{h}_d^{(T)}) = LSTM_d(\mathbf{v}_d^{(1)}, \mathbf{v}_d^{(2)}, ..., \mathbf{v}_d^{(T)}) \tag{15}$$

$$p(\widehat{y_d}|\mathbf{x}) = Sigmoid\left(\frac{1}{T}(\tanh{(\boldsymbol{h}_d^{(1)})} \odot \mathbf{v}^{(1)} + \tanh{(\boldsymbol{h}_d^{(2)})} \odot \mathbf{v}^{(2)} + ... + \tanh{(\boldsymbol{h}_d^{(T)})} \odot \mathbf{v}^{(T)})\mathbf{W}_d^o + b_d^o\right) \tag{16}$$

**4) MTL-LSTM.** The naive hard-sharing multi-task learning method where all tasks share the same network except for the separate output layers for prediction, whose base network is Long Short-Term Memory Network (LSTM).

$$(\mathbf{v}^{(1)}, \mathbf{v}^{(2)}, ..., \mathbf{v}^{(T)}) = \mathbf{v} = \mathbf{x}\mathbf{W}_{emb} \in \mathbb{R}^{T \times k} \tag{17}$$

$$(\mathbf{h}^{(1)}, \mathbf{h}^{(2)}, ..., \mathbf{h}^{(T)}) = LSTM(\mathbf{v}^{(1)}, \mathbf{v}^{(2)}, ..., \mathbf{v}^{(T)}) \tag{18}$$

$$p(\widehat{y_d}|\mathbf{x}) = Sigmoid\left(\frac{1}{T}(\tanh{(\boldsymbol{h}^{(1)})} \odot \mathbf{v}^{(1)} + \tanh{(\boldsymbol{h}^{(2)})} \odot \mathbf{v}^{(2)} + ... + \tanh{(\boldsymbol{h}^{(T)})} \odot \mathbf{v}^{(T)})\mathbf{W}_d^o + b_d^o\right) \tag{19}$$

**5) MTL-Transformer.** The same as MTL-LSTM, but with Transformer (Vaswani et al., 2017) as the base network.

$$\mathbf{v} = \mathbf{x}\mathbf{W}_{emb} + POS\_ENC \in \mathbb{R}^{T \times k} \tag{20}$$

$$\mathbf{f} = TRANS\_BLOCK(v) \in \mathbb{R}^{T \times k} \tag{21}$$

$$\mathbf{c}_i = \frac{1}{T}(\mathbf{f}^{(1)} + \mathbf{f}^{(2)} + ... + \mathbf{f}^{(T)}) \tag{22}$$

$$p(\widehat{y_d}|\mathbf{x}) = Sigmoid\,(\mathbf{c}_i\mathbf{W}_d^o + b_d^o) \tag{23}$$

where $POS\_ENC$ is the positional encoding used in Transformer, $TRANS\_BLOCK$ is also the architechture used in the paper, which consists of 2 sublayers: $MULTI\_HEAD$ (with 4 heads) and $FFW$. We also used residual connection and layer norm after each sublayer as the original paper.

**6) MTL-RETAIN.** The same as MTL-LSTM, but with RETAIN (Choi et al., 2016) as the base network. Specifically, after getting the shared context vector $\mathbf{c}_i$, separated output layers will be applied to form the prediction for each task.

$$\mathbf{c}_i : \text{context vector from RETAIN} \tag{24}$$

$$p(\widehat{y_d}|\mathbf{x}) = Sigmoid\,(\mathbf{c}_i\mathbf{W}_d^o + b_d^o) \tag{25}$$

**7) MTL-UA.** The same as MTL-LSTM, but with UA (Heo et al., 2018) as the base network. Specifically, after getting the shared context vector $\mathbf{c}_i$, separated output layers will be applied to form the prediction for each task. This can be seen as the probabilistic version of MTL-RETAIN.

$$\mathbf{c}_i : \text{context vector from UA} \tag{26}$$

$$p(\widehat{y_d}|\mathbf{x}) = Sigmoid\,(\mathbf{c}_i\mathbf{W}_d^o + b_d^o) \tag{27}$$

**8) AMTL-LSTM.** This is asymmetric multi-task learning (Lee et al., 2016) adopted for our time-series prediction framework, where we learn the knowledge transfer graph between task-specific parameters, which is learned to perform asymmetric knowledge transfer based on the task loss. Since the parameters for each task is shared across all timesteps, this will result in static asymmetric transfer between tasks.

**9) MTL-RETAIN-Kendall** This is the work proposed by (Kendall et al., 2018). The base model we use is MTL-RETAIN. However, when combining the losses, we followed the work by (Kendall et al., 2018):

$$\sum_{d=1}^{D}\left(\frac{1}{\sigma_d^2}L_d + log(\sigma_d)\right) \tag{28}$$

**10) TP-AMTL.** Our probabilistic temporal asymmetric multi-task learning model that performs both intra-task and inter-task knowledge transfer.

## A.3 Details of models in Ablation study

**AMTL-intratask.** $\mathbf{C}_d^{(t)} = \mathbf{f}_d^{(t)} + \sum_{i=1}^{t} \alpha_{d,d}^{(i,t)} * G_d\left(\mathbf{f}_d^{(i)}\right) \forall t \in \{1, 2, ..., T\}$

**AMTL-samestep.** $\mathbf{C}_d^{(t)} = \mathbf{f}_d^{(t)} + \sum_{j=1}^{D} G_{j,d}\left(\mathbf{f}_j^{(t)}\right) \forall t \in \{1, 2, ..., T\}$

**TD-TAMTL.** The deterministic version of our model that does not make use of feature-level uncertainty when performing knowledge transfer.

$$\mathbf{v} = \mathbf{x}\mathbf{W}_{emb} \in \mathbb{R}^{T \times k}$$

$$\mathbf{h} = (\mathbf{h}^{(1)}, \mathbf{h}^{(2)}, ..., \mathbf{h}^{(T)}) = RNN(\mathbf{v}^{(1)}, \mathbf{v}^{(2)}, ..., \mathbf{v}^{(T)})$$

$$\mathbf{h}_d = \sigma((...\sigma(\sigma(\mathbf{h}\mathbf{W}_d^1 + \mathbf{b}_d^1)\mathbf{W}_d^2 + \mathbf{b}_d^2)...)\mathbf{W}_d^L + \mathbf{b}_d^L) \in \mathbb{R}^{T \times k}$$

$$(\mathbf{f}_d^{(1)}, \mathbf{f}_d^{(2)}, ..., \mathbf{f}_d^{(T)}) = \mathbf{h}_d$$

$$\mathbf{C}_d^{(t)} = \mathbf{f}_d^{(t)} + \sum_{j=1}^{D}\sum_{i=1}^{t} \alpha_{j,d}^{(i,t)} * G\left(\mathbf{f}_j^{(i)}\right) \forall t \in \{1, 2, ..., T\}$$

$$\boldsymbol{\beta}_d^{(t)} = tanh\left(\mathbf{C}_d^{(t)}\mathbf{W}_d^\beta + \mathbf{b}_d^\beta\right) \quad \forall t \in \{1, 2, ..., T\}$$

$$p(\widehat{y_d}|\mathbf{x}) = Sigmoid\left(\frac{1}{T}(\boldsymbol{\beta}_d^{(1)} \odot \mathbf{v}^{(1)} + \boldsymbol{\beta}_d^{(2)} \odot \mathbf{v}^{(2)} + ... + \boldsymbol{\beta}_d^{(T)} \odot \mathbf{v}^{(T)})\mathbf{W}_d^o + b_d^o\right)$$

where $\alpha_{j,d}^{(i,t)} = F_\theta(\mathbf{f}_f^{(i)}, \mathbf{f}_d^{(t)})$

**TP-TAMTL - no constraints.** $\mathbf{C}_d^{(t)} = \mathbf{f}_d^{(t)} + \sum_{j=1}^{D}\sum_{i=1}^{T} \alpha_{j,d}^{(i,t)} * G_{j,d}\left(\mathbf{f}_j^{(i)}\right) \forall t \in \{1, 2, ..., T\}$

## A.4 Configuration and parameters

We trained all the models using Adam optimizer with dropout regularization. We set the maximum iteration for Adam optimizer as 100,000, and for other hyperparameters, we searched for the optimal values by cross-validation, within predefined ranges as follows: Hidden units: {8, 16, 32, 64}, number of layers: {2,3,6}, mini batch size: {32, 64, 128, 256}, learning rate: {0.01, 0.001, 0.0001},*L2* regularization: {0.02, 0.002, 0.0002,0.00}, and dropout rate {0.1, 0.15, 0.2, 0.25, 0.3, 0.4, 0.5}.

## B Quantitative evaluation on clinical time-series prediction task : MIMIC III-Heart Failure

Here, we provide the experimental results of our model and other baselines on the additional dataset: MIMIC III-Heart Failre. Table 5 shows that our model still outperforms other baselines, which indicates that our method can generalize well on a variaty of time-series datasets.

## C Clinical Interpretation of generated uncertainty and knowledge transfer between tasks

In this section, we further describe the interpretation of several example patients using generated uncertainty and knowledge transfer across timesteps.

The example patient in Figure 5 had a fever on admission and was confirmed to be positive with bacterial infection on culture study in this specific timestep 2:57. The patient continued to have

Table 5: MIMIC-III Heart Failure

| Models | | Tasks | | | | |
|---|---|---|---|---|---|---|
| | | Ischemic | Valvular | Heart Failure | Mortality | Average |
| STL | LSTM | 0.7072± 0.01 | 0.7700± 0.02 | 0.6899± 0.02 | 0.7169± 0.03 | 0.7210± 0.01 |
| | RETAIN (Choi et al., 2016) | 0.6573± 0.03 | 0.7875± 0.01 | 0.6850± 0.01 | 0.7027± 0.02 | 0.7081± 0.01 |
| | UA (Heo et al., 2018) | 0.6843± 0.01 | 0.7728± 0.02 | **0.7090± 0.01** | **0.7191± 0.01** | 0.7213± 0.01 |
| MTL | LSTM | 0.6838± 0.02 | 0.7808± 0.02 | 0.6965± 0.01 | 0.7093± 0.02 | 0.7254± 0.02 |
| | TRANS (Vaswani et al., 2017) | 0.6801± 0.01 | 0.7693± 0.01 | **0.7098± 0.02** | 0.7008± 0.02 | 0.7150± 0.02 |
| | RETAIN | 0.6649± 0.01 | 0.7532 ± 0.03 | 0.6868± 0.02 | 0.7023± 0.03 | 0.7018± 0.02 |
| | UA | 0.6917± 0.01 | 0.7868± 0.01 | 0.7073± 0.01 | 0.7029± 0.01 | 0.7222 ± 0.01 |
| | RETAIN-Kendall | 0.6476± 0.03 | 0.7712± 0.02 | 0.6826± 0.01 | 0.7017± 0.02 | 0.7008± 0.01 |
| | AMTL-LSTM (Lee et al., 2016) | 0.6963± 0.01 | **0.7997± 0.02** | 0.7006± 0.01 | 0.7108± 0.01 | 0.7268± 0.01 |
| | TP-AMTL (our model) | **0.7113± 0.01** | **0.7979± 0.01** | **0.7103± 0.01** | **0.7185± 0.02** | **0.7345± 0.01** |

fever, and white blood cell count increased to the state of leukocytosis. Also, both systolic and diastolic blood pressure declined over time. We can see that uncertainty of target task drops when the model can confidently infer to the patient status from feature values, and aids from source target can decrease in that case. Figure 5a represent knowledge transfer from one timestep of a source task to multiple time steps in the target task. We examine how multiple time steps of the source task transfers knowledge to certain time steps in the target task from the same example patient of MIMIC-III Infection dataset used in Figure 5a. On the vicinity of the same timepoint where this patient was confirmed to have bacterial infection, we can see that the uncertainty of source target starts to increase, and knowledge reversely flows to source task *fever*. This happens in accordance with the drop of knowledge transfer from *fever* to *infection* in Figure 5a. We can infer that the knowledge from task *infection* becomes more useful to predict source *fever* in this timestep as patient condition related to this task is happening around this time step.

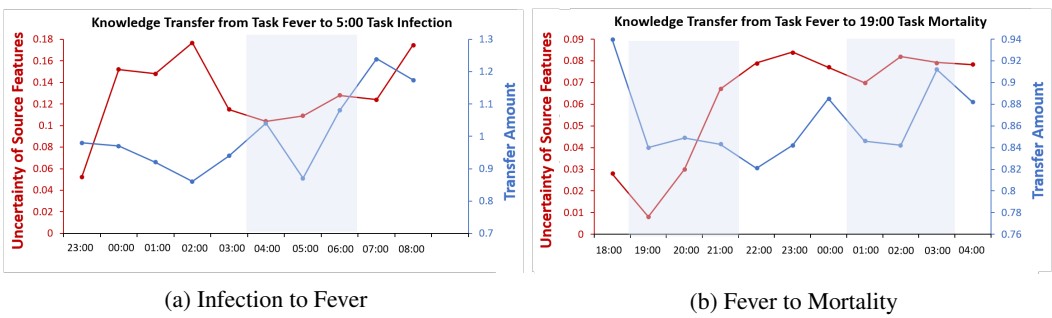

(a) Infection to Fever          (b) Fever to Mortality

Figure 7: **Visualizations of the amount of uncertainty and knowledge transfer for example cases** The changes in the amount of uncertainty at certain timesteps are correlated with clinical events. We denote the timesteps with noticeable changes in uncertainty and knowledge transfer with blue boxes.

Additionally, we select other example patient from MIMIC-III Infection dataset (Figure 7a 7b). This patient had fever and leukocytosis (elevation of white blood cell as a result of bacterial intrusion, implies infectious status of a patient) at the earlier timepoint of admission but was not confirmed to have infection afterwards. The first timepoint highlighted with the blue box is when this patient had fever and started to recover from fever. At 19:00 when this patient had fever, the uncertainty of source task *fever* decreases and this task transfers more to target task *mortality*. As the patient recovers from fever, the uncertainty of task fever increases and knowledge transfer from fever to mortality drops accordingly. The second blue box on the right denotes the time step when the complete blood count lab result showed this patient has leukocytosis, which implies the high propensity of infection. Knowledge transfer starts to drop as the knowledge from source task *fever* is less important as the uncertainty of target task *mortality* drops.

Interpretation on another example patient from MIMIC III-heart failure dataset is plotted on Figure 8. This example patient is finally diagnosed with congestive heart failure on Chest X-ray. During admission period, troponin level of this patient was elevated, which is not diagnostic (Reichlin et al., 2009), but implying that this patient had cardiac event. Given cardiac events, hypotension occured in 1:21 (Table **??**, Figure 8) can be explained to be related to final diagnosis heart failure. As

Table 6: Clinical Events in selected medical records for case studies. **MechVent** - Mechanical Ventilation, **FiO2** - Fractional inspired Oxygen, **SBP** - Systolic arterial blood pressure, **DBP** - Diastolic arterial blood pressure, **HR** - Heart Rate, **Temp** - Body Temperature, **Urine** - Urine output, **GCS** - Glasgow Coma Score, **WBC** - White Blood Cell Count, **Culture** - Culture Results.

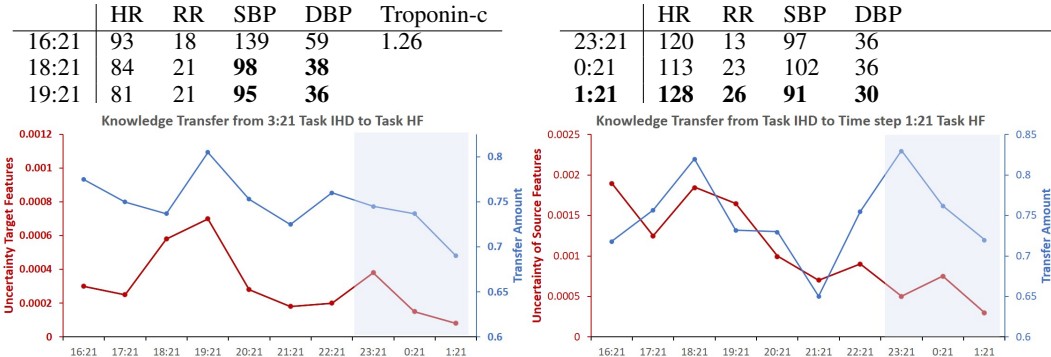

|  | HR | RR | SBP | DBP | Troponin-c |
|---|---|---|---|---|---|
| 16:21 | 93 | 18 | 139 | 59 | 1.26 |
| 18:21 | 84 | 21 | **98** | **38** | |
| 19:21 | 81 | 21 | **95** | **36** | |

|  | HR | RR | SBP | DBP |
|---|---|---|---|---|
| 23:21 | 120 | 13 | 97 | 36 |
| 0:21 | 113 | 23 | 102 | 36 |
| **1:21** | **128** | **26** | **91** | **30** |

(a) KT from single timestep and Target Uncertainty     (b) KT to single timestep and Source Uncertainty

Figure 8: **Uncertainty and Knowledge Transfer(KT) : Example case of MIMIC III - Heart Failure dataset** where the changes in the amount of uncertainty at certain timesteps are correlated with clinical events. We denote the timesteps with noticeable changes in uncertainty and knowledge transfer with blue boxes.

the patient's SBP decreases to 90 and DBP to 30 around 1:21 (Table **??**), the uncertainty of target task *Heart Failure* decreases in Figure 8a. Knowledge transfer starts to drop as the knowledge from target task becomes more important than that of source task. We can also see that the trend of knowledge transfer follows the trend of target uncertainty. Furthermore, troponin increased in 16:21 implies ongoing myocardial stress, which can be expressed as constantly lowered uncertainty of source task *ischemic heart disease* among the window period we plotted on Figure 8b. As uncertainty of source task decreases, knowledge transfer to target task *heart failure* kept increasing till 23:21. However, as patient condition related to heart function, especially blood pressure starts to decrease and knowledge from target task gets important, knowledge transfer starts to decrease after 23:21.

Table 7: Feature information of MIMIC III - Infection dataset

| Features | Item-ID | Name of Item |
|---|---|---|
| Age | NA | initime
dob |
| Sex | NA | gender |
| Heart Rate | 211
22045 | Heart Rate
Heart Rate |
| Systolic Blood Pressure | 51
442
455
6701
220179
220050 | Systolic Blood Pressure
Systolic Blood Pressure
Systolic Blood Pressure
Systolic Blood Pressure
Systolic Blood Pressure
Systolic Blood Pressure |
| Diastolic Blood Pressure | 8368
8440
8441
8555
220051
220180 | Diastolic Blood Pressure
Diastolic Blood Pressure
Diastolic Blood Pressure
Diastolic Blood Pressure
Diastolic Blood Pressure
Diastolic Blood Pressure |
| Glasgow Coma Scale | 223900
223901
220739 | GCS-Verbal Response
GCS-Motor Response
GCS-Eye Opening |
| Invasive procedures | 225433
5456
225445
225446
225399
5939
225469
225442
224264
224560
225430
225315
226475
5889 | Chest Tube Placed
Chest Tube
Paracentesis
PEG Insertion
Lumbar Puncture
Lumbar drain
OR Received
Liver Biopsy
PICC Line
PA Catheter
Cardiac Cath
Tunneled (Hickman) Line
Intraventricular Drain Inserted
Bladder cath |
| Endoscopic Procedure | 225434
225439
227550 | Colonoscopy
Endoscopy
ERCP |
| Intubation / Unplanned Extubation | 224385
225448
225468
225477
226237
225792 | Intubation
Percutaneous Tracheostomy
Unplanned Extubation (patient-initiated)
Unplanned Extubation (non-patient initiated)
Open Tracheostomy
Invasive Ventilation |
| Albumin | 772
1521
227456
3727
226981
226982 | Albumin (>3.2)
Albumin
Albumin
Albumin (3.9-4.8)
Albumin_ApacheIV
AlbuminScore_ApacheIV |
| Total Protein | 220650
849
3807
1539
220650 | Total Protein(6.5-8)
Total Protein(6.5-8)
Total Protein
Total Protein(6.5-8)
Total Protein(6.5-8) |

| Features | Item-ID | Name of Item |
|---|---|---|
| Dialysis | 225441 | Hemodialysis |
| | 225805 | Peritoneal Dialysis |
| | 226477 | Temporary Pacemaker Wires Inserted |
| | 224270 | Dialysis Catheter |
| | 225802 | Dialysis - CRRT |
| | 225805 | Peritoneal Dialysis |
| Intravenous Steroid | 4929 | Prednisolone |
| | 7772 | Predisolone |
| | 6753 | Prednisilone gtts |
| | 6111 | prednisone |
| | 8309 | prednisolone gtts |
| | 5003 | prednisolone |
| | 1878 | methylprednisolone |
| | 2656 | SOLUMEDROL MG/KG/HR |
| | 2657 | SOLUMEDROL CC/H |
| | 2629 | SOLUMEDROL DRIP |
| | 2983 | solumedrol mg/hr |
| | 7425 | Solu-medrol mg/hr |
| | 6323 | solumedol |
| | 7592 | Solumedrol cc/h |
| | 30069 | Solumedrol |
| | 2959 | Solumedrolmg/kg/hr |
| | 1878 | methylprednisolone |
| | 5395 | Beclamethasone |
| | 4542 | Tobradex |
| | 5612 | Dexamethasone gtts |
| | 3463 | Hydrocortisone |
| | 8070 | dexamethasone gtts |

Table 8: Feature information of MIMIC III - Heart Failure dataset

| Features | Item-ID | Name of Item |
|---|---|---|
| Age | NA | initime
dob |
| Sex | NA | gender |
| Heart Rate | 211 | Heart Rate |
| | 22045 | Heart Rate |
| Respiratory Rate | 618 | Respiratory Rate |
| | 619 | Respiratory Rate |
| | 220210 | Respiratory Rate |
| | 224688 | Respiratory Rate |
| | 224689 | Respiratory Rate |
| | 224690 | Respiratory Rate |
| Systolic Blood Pressure | 51 | Systolic Blood Pressure |
| | 442 | Systolic Blood Pressure |
| | 455 | Systolic Blood Pressure |
| | 6701 | Systolic Blood Pressure |
| | 220179 | Systolic Blood Pressure |
| | 220050 | Systolic Blood Pressure |
| Diastolic Blood Pressure | 8368 | Diastolic Blood Pressure |
| | 8440 | Diastolic Blood Pressure |
| | 8441 | Diastolic Blood Pressure |
| | 8555 | Diastolic Blood Pressure |
| | 220051 | Diastolic Blood Pressure |
| | 220180 | Diastolic Blood Pressure |
| Body Temperature | 676 | Body Temperature |
| | 677 | Body Temperature |
| | 8537 | Body Temperature |
| | 223762 | Body Temperature |
| | 226329 | Body Temperature |
| Fraction of inspired oxygen (Fi$O_2$) | 189 | Fi$O_2$ |
| | 190 | Fi$O_2$ |
| | 2981 | Fi$O_2$ |
| | 3420 | Fi$O_2$ |
| | 3422 | Fi$O_2$ |
| | 223835 | Fi$O_2$ |
| Mixed venous Oxygen Saturation ($S_vO_2$) | 823 | $S_vO_2$ |
| | 2396 | $S_vO_2$ |
| | 2398 | $S_vO_2$ |
| | 2574 | $S_vO_2$ |
| | 2842 | $S_vO_2$ |
| | 2933 | $S_vO_2$ |
| | 2955 | $S_vO_2$ |
| | 3776 | $S_vO_2$ |
| | 5636 | $S_vO_2$ |
| | 6024 | $S_vO_2$ |
| | 7260 | $S_vO_2$ |
| | 7063 | $S_vO_2$ |
| | 7293 | $S_vO_2$ |
| | 226541 | $S_vO_2$ |
| | 227685 | $S_vO_2$ |
| | 225674 | $S_vO_2$ |
| | 227686 | $S_vO_2$ |
| Oxygen Saturation of arterial blood ($S_aO_2$) | 834 | $S_aO_2$ |
| | 3288 | $S_aO_2$ |
| | 8498 | $S_aO_2$ |
| | 220227 | $S_aO_2$ |
| Brain Natriuretic Peptide (BNP) | 7294 | BNP |
| | 227446 | BNP |
| | 225622 | BNP |
| Ejection Fraction (EF) | 227008 | EF |
| Glasgow Coma Scale (GCS) - Verbal Response | 223900 | GCS-Verbal Response |
| Glasgow Coma Scale (GCS) - Motor Response | 223901 | GCS-Motor Response |
| Glasgow Coma Scale (GCS) - Eye Opening | 220739 | GCS-Eye Opening |

