# OpenReview forum: "Temporal Probabilistic Asymmetric Multi-task Learning"
_ICLR.cc/2020/Conference — Reject_

### Official Review · AnonReviewer2 · 2019-10-22
**Official Blind Review #2**

**Rating:** 6

**Review:**

The submission proposes a probabilistic method for multitask learning for sequence data focusing on the use of uncertainty to control the amount of transfer between tasks. In comparison to prior work, the approach directly models uncertainty by parametrising Gaussian distributions and dropout variational inference. The approach additionally models transfer between features from different time-steps. The evaluation shows that the method outperforms recent state-of-the-art models on two medical benchmarks.

The paper is overall well written but includes some vagueness regarding the proposed method as well as the experimental setup. While describing techniques to model two kinds of uncertainty the aspect of the model which adapts transfer to the amount of uncertainty in source and target features is not described in detail (F_theta). It is described as if it only received samples for both features without any additional information regarding uncertainty. This makes it improbable that the model is able to take the feature’s uncertainty into account when modelling transfer.

The paper follows in section 3.3 with an argument for the lower computational complexity of only transferring from previous time-steps. I might misunderstand the description but having every time-step’s features be influenced only by previous timesteps should still result in complexity O(T^2) (even though the amount of computation is reduced by a constant factor).

My background is not in machine learning with clinical data (so I do not know common datasets) but it is unclear to me why the datasets evaluated on only consist of a very small section of the overall dataset they’re taken from (in case 1). Looking at table 4 the main factor seems to be probabilistic modelling so additional ablations with either version of uncertainty would be interesting.

By introducing probabilistic modelling to control the amount of transfer between tasks the paper provides an interesting perspective and is able to show strong performance. However it is also quite vague on important aspects such as the exact modelling of transfer and limited regarding evaluation.

Minor aspects:
- V from the final equation on page 5 is never described.
- After describing modelling both kinds of uncertainty in the method section, the reader is uninformed which uncertainty is shown in plots ⅘
- Table 4 does not have the two highest results in bold (see AMTL samestep)
- Very limited multitask and transfer learning references pre 2016.
- Some descriptions are unclear including the mention of ‘attention allocation’


**Experience Assessment:**

I have read many papers in this area.

**Review Assessment: Checking Correctness Of Derivations And Theory:**

I assessed the sensibility of the derivations and theory.

**Review Assessment: Checking Correctness Of Experiments:**

I assessed the sensibility of the experiments.

**Review Assessment: Thoroughness In Paper Reading:**

I read the paper at least twice and used my best judgement in assessing the paper.

---

> ### Author Response · Authors · 2019-11-11
> **Response to Reviewer #2**
>
> We thank you for your constructive comments.
>
> 1) The aspect of the model which adapts transfer to the amount of uncertainty in source and target features is not described in detail ($F_\theta$). It is described as if it only received samples for both features without any additional information regarding uncertainty. This makes it improbable that the model is able to take the feature’s uncertainty into account when modelling transfer.
>
> - The network $F_\theta$ indeed only takes the instances of the distributions as input. However, note that the mean and variance of the distribution are closely correlated (as they are output of a network). Therefore, by feeding the instances (which are closely related to the means), we expect the network to infer the correlation to the feature’s uncertainty into account. We also empirically showed that this works well in practice, since the amount of knowledge transfer correlated to the features’ uncertainty to some degree (See Figure 4 and 5).
>
> 2) Having every time-step’s features be influenced only by previous timesteps should still result in complexity $O(T^2)$ (even though the amount of computation is reduced by a constant factor).
>
> - We apologize for the confusion. The complexity analysis is for single model update at the inference time. When we receive new data for a patient, we only need to compute its task-specific features and perform knowledge transfer from previous time step to it ($O(T)$). Note that we don’t need to update the knowledge for the past time-step, since there is no future-to-past transfer. The total training and inference complexity would be $O(T^2)$ as you mentioned. We have clarified this in the revision.
>
> 3) It is unclear to me why the datasets evaluated on only consist of a very small section of the overall dataset they’re taken from (in case 1).
>
> - The size of Physionet2012challenge dataset is 8000, but only 4000 of them have labels (the other 4000 are without labels and were used as test set in the challenge). Therefore, we used 4000 data instances with labels. Regarding the MIMIC-III dataset, it is a very sparse dataset. Some patients do not even have (or have very few) records for the collected features in the interested time window. Therefore, we discarded these cases and only collected instances with sufficient amount of features. Also, we use the first 48 hours after admission for each patient to only consider the patient's condition on admission, as infection occurring after 48 hours is more likely to be acquired at the ICUs. We have clarified this point in the revision.
>
>
> 4) Looking at table 4 the main factor seems to be probabilistic modelling so additional ablations with either version of uncertainty would be interesting.
>
> - Thank you so much for the helpful suggestion. In the revision, we provide the ablation study for 2 kinds of uncertainty in the Physionet2012 dataset in Table 4.
>
> From this ablation study, We can see that epistemic uncertainty attributes more to the performance gain (0.8694 with the epistemic uncertainty and 0.8254 with the aleatoric uncertainty, where the base model performance is 0.8133). However, it should be noted that the impact of two kinds of uncertainty may largely vary from a dataset to dataset, and thus the best solution is to model them both, as in our temporal asymmetric MTL framework.
>
> 5) By introducing probabilistic modelling to control the amount of transfer between tasks the paper provides an interesting perspective and is able to show strong performance. However it is also quite vague on important aspects such as the exact modelling of transfer and limited regarding evaluation.
>
> - We apologize for the lack of details for network F. We omitted the description on F due to space limitation, but added it back into the revision. However, other implementation works well too (that’s why we describe it in a general form).

---

### Official Review · AnonReviewer3 · 2019-10-23
**Official Blind Review #3**

**Rating:** 6

**Review:**

In this paper, the authors proposed an asymmetric multi-task learning method on temporal data, and applied it to clinical risk prediction.

Overall, the problem studied in this paper is interesting, the proposed method looks reasonable, and the application is also interesting. However, there are also some concerns.

1. The details on how to construct F_\theta to generate the hyper-parameter of a in (10) are missing. As the proposed MTL method is asymmetric, F_\theta(f, g) is supposed to be different from F_\theta(g, f), right? Otherwise, the transfer weight \alpha from f to g would be the same as the one from g to f. What is the design of F_\theta.  Moreover, the authors mentioned that "the network F can learn a meaningful distribution of the transfer weight, such that it sets the value of \alpha high when f and g are related, with low uncertainty on f and high uncertainty on g." However, only based on (8), (9) and (10), it is quite difficult to understand why the claim can be implemented. More details are needed.

2. The complexity analysis is only focused on the relation to the number of timestamps T, without taking the number of tasks D into consideration. As the proposed method is asymmetric across tasks and across time, when the number of tasks is large, the scalability may be an issue. In real-world applications, e.g., in a big hospital, the number of patients can be very large. In this case, is the proposed method practical?

3. Though the proposed method looks reasonable, it contains many components or networks. I guess to train such a composition network precisely needs a lot of tricks in practice.


**Experience Assessment:**

I have published one or two papers in this area.

**Review Assessment: Checking Correctness Of Derivations And Theory:**

N/A

**Review Assessment: Checking Correctness Of Experiments:**

I assessed the sensibility of the experiments.

**Review Assessment: Thoroughness In Paper Reading:**

I read the paper at least twice and used my best judgement in assessing the paper.

---

> ### Author Response · Authors · 2019-11-11
> **Response to Reviewer #3**
>
> We thank you for your constructive comments. We respond to your comments below:
>
> 1.1) The details on how to construct F_\theta to generate the hyper-parameter of a in (10) are missing. As the proposed MTL method is asymmetric, F_\theta(f, g) is supposed to be different from F_\theta(g, f), right? Otherwise, the transfer weight \alpha from f to g would be the same as the one from g to f. What is the design of F_\theta.
>
> - We apologize for the confusion. We removed the detailed descriptions of F_\theta due to page limit. In general, $F_\theta$ should be a function from $R^{2k}$ to $R^2$, and we adopt a simple implementation as follows: $\mu_a=\sigma(\sigma(f W_f+b_f) \cdot \sigma(g W_g+b_g))$ and $\sigma_a=softplus(\mu_a w+b)$, where $\sigma$ is some activation function (which should not be confused with $\sigma_a$) and $\cdot$ is the inner product. We included the descriptions back into the revision.
>
> 1.2) Moreover, the authors mentioned that "the network F can learn a meaningful distribution of the transfer weight, such that it sets the value of \alpha high when f and g are related, with low uncertainty on f and high uncertainty on g." However, only based on (8), (9) and (10), it is quite difficult to understand why the claim can be implemented. More details are needed.
>
> - Equation (10) shows that $F_\theta$ only takes instances of the distribution of f and g as input. However, since the features are probabilistic due to MC dropout and conditional distributions, the network G will learn to assign small weights to features that vary largely due to high variance, and assign high weights to features that do not change as much due to small variance. In the extreme case, if the noise is very large, the feature will obtain weight close to 0. We empirically show in Figure 4 and 5 that the amount of knowledge transfer is strongly correlated with the feature-level uncertainty.
>
>
> 2) The complexity analysis is only focused on the relation to the number of timestamps T, without taking the number of tasks D into consideration. As the proposed method is asymmetric across tasks and across time, when the number of tasks is large, the scalability may be an issue. In real-world applications, e.g., in a big hospital, the number of patients can be very large. In this case, is the proposed method practical?
>
> - Our model’s training complexity would scale quadratically with the number of tasks (O(D^2)). However this is a common problem with any multi-task learning framework with task-to-task relationship modeling. Further, regarding the number of patients in large hospitals, we could treat the record of each patient as a single instance, and consider each task as a different clinical prediction task (fever, infection or mortality) as we did in our experiments. In that case, the scalability to large number of instances is no longer a serious problem as we will use stochastic gradient descent to tackle the scalability issue.
>
>
> 3) Though the proposed method looks reasonable, it contains many components or networks. I guess to train such a composition network precisely needs a lot of tricks in practice.
>
> - Our model does not require a lot of tricks or hyperparameter tuning. The only additional part of this model to naive layer-sharing MTL framework is the small knowledge transfer network G_\phi, and the model does not require anything other than performing MC-dropout at test time. We provided the codes of the model and other baselines for reproduction, which is simple to use and re-implement.

---

### Official Review · AnonReviewer1 · 2019-10-27
**Official Blind Review #1**

**Rating:** 3

**Review:**

This paper proposes a temporal probabilistic asymmetric multi-task learning model for sequential data. The asymmetric property lies in the feature level.

This paper criticize that the task loss used in previous works may be unreliable as a measure of the knowledge from the task. However, the feature uncertainty used in the proposed model can also be unreliable and I cannot see any reliable guarantee.

The organization of Section 3 is not good. The logics in different subsections are not so clear. Some notations seem undefined.

Eq. (4) defines p(z_d|x,\omega). But later it defines z_d is from p(z_d|x,y_d,\omega). The difference between these two distributions lies in y_d. I don’t know which one is used. What is p_\theta(z_d|x,\omega)? The notations need to be properly defined.

Based on Section 3.3, the proposed model seems to require different tasks have the same total time step T and this requirement is a bit strong.

It seems that there is no Figure 2c.

**Experience Assessment:**

I have published in this field for several years.

**Review Assessment: Checking Correctness Of Derivations And Theory:**

I assessed the sensibility of the derivations and theory.

**Review Assessment: Checking Correctness Of Experiments:**

I assessed the sensibility of the experiments.

**Review Assessment: Thoroughness In Paper Reading:**

I read the paper at least twice and used my best judgement in assessing the paper.

---

> ### Author Response · Authors · 2019-11-11
> **Response to Reviewer 1**
>
> We thank you for your constructive comments. During the rebuttal period, we did our best to respond to your
>
> 1) This paper criticize that the task loss used in previous works may be unreliable as a measure of the knowledge from the task. However, the feature uncertainty used in the proposed model can also be unreliable and I cannot see any reliable guarantee.
>
> - This is a critical misunderstanding. The foremost reason we use feature-level uncertainty as a measure of knowledge reliability is because the task loss may not be available at every timestep (line 5, page 2) for time-series prediction task, as the prediction is often made at the last timestep. This makes it impossible to use task loss to transfer knowledge from one timestep to another. On the contrary, using feature-level uncertainty makes it possible to measure the reliability of knowledge at each timestep.
>
> Secondly, we do not claim that feature-level uncertainty is the perfect measure of knowledge reliability. Rather, we use it as a more direct proxy of reliability over task loss. As mentioned in line 3, page 2, if the model is trained with few instances, the loss could be easily reduced, but it does not mean that the model is reliable as the learned knowledge may be specific to the few instances and may not generalize. Feature-level uncertainty, on the other hand, does not exhibit such overfitting behavior. Further, the experimental results on PhysioNet 2012 confirms the superiority of feature-level uncertainty over loss in asymmetric multi-task learning framework. The results show that our model with the feature-level uncertainty without any across-timestep transfer (AMTL-samestep, Table 4) performs significantly better (0.8557) than its loss-based asymmetric MTL counterpart, AMTL-LSTM (0.8022). The full model with across-timestep transfer achieves the best accuracy (0.8719).
>
>
> 2) The organization of Section 3 is not good. The logics in different subsections are not so clear. Some notations seem undefined.
>
> - We organized Section 3 to first talk about the base model architecture (subsection 3.1), and then describe the probabilistic knowledge transfer mechanism (subsection 3.2) and knowledge transfer in time-series prediction tasks (subsection 3.3). We find it sufficiently clear, but we will reflect your suggestion if you provide us more detailed comments on how to improve the organization and notation.
>
>
> 3) Eq. (4) defines $p(z_d|x,\omega)$. But later it defines z_d is from $p(z_d|x,y_d,\omega)$. The difference between these two distributions lies in y_d. I don’t know which one is used. What is $p_\theta(z_d|x,\omega)$? The notations need to be properly defined.
>
> - This is a misunderstanding, and there is no typo in the notations. We did not use the notation $p(z_d|x,\omega)$, and what we have there instead is $p_\theta(z_d|x,\omega)$, which is the (conditional) prior of $z_d$ obtained from the prior network parameterized by $\theta$. $p(z_d|x, y_d, \omega)$ is not a typo but is the posterior of $z_d$. We did not explicitly mention them as prior and posterior in the paper as they can be clearly identified as the prior and the posterior in the context of Bayesian deep learning. However, we added in explicit descriptions of them in the revision for improved clarity.
>
>
> 4) The proposed model seems to require different tasks have the same total time step T and this requirement is a bit strong.
>
> - Our multi-task learning framework does not require different tasks to have the same number of timesteps. Actually, the number of timesteps differs even from instance to instance. We simply use zero-padding to deal with this discrepancy, which is a standard procedure for prediction with time-series data. In the revision, we made it clear that the number of timesteps for each task and instance could be different.
>
>
> 5) It seems that there is no Figure 2c.
>
> - There is Figure 2(c). Figure 2(c) illustrates task-specific latent features. In the revision, we edited the figure such that Figure 2(c) cannot be missed.

---

### Author Response · Authors · 2019-11-11
**Summary of the revision**

During the rebuttal period, we have made the following changes to the paper, based on the reviewers' comments.

-    We modified Figure 2 such that Figure 2(c) cannot be mistaken as missing.
-    We modified the texts in variational inference part to explicitly describe the posterior and the prior (section 3.1).
-    We clarified the model complexity (section 3.3) and the dataset size (section4.1).
-    We added in references suggested by the reviewers, such as (Argyriou et al., 2008), (Yang & Hospedales, 2016a), (Yang & Hospedales, 2016b), (Ruder12 et al.), (Kang et al., 2011), (Kumar & Daume III, 2012) and (Maurer et al., 2013) (section2).
-    We added in details about network F_\theta(f,g) which outputs the knowledge transfer weight (section 3.2)
-    We clarified that v from Eq.13 is the shared embedding from Eq.1 (section 3.3)
-    We added the ablation study for two kinds of uncertainty: epistemic and aleatoric in Table 4 (section 4.5)
-    We modified Table 4 to correctly highlight the best performing models (section 4.5).
-    We have corrected typos in the appendix (Table 6)

---

### Decision · Program_Chairs · 2019-12-19

**Decision:**

Reject

**Comment:**

The authors propose a method for multi-task learning with time series data. The reviewers found the paper interesting, but the majority found the description of the method in the paper confusing and several technical details missing. Moreover, the reviewers were not convinced that the technique used for uncertainty quantification of the features at each stage of the time series is well founded.